



# Hydrocarbon accumulation in basins with multiple phases of extension and inversion: examples from the Western Desert (Egypt) and the Western Black Sea

William Bosworth[1], Gábor Tari[2]

[1] Apache Egypt Companies, 11 Street 281, New Maadi, Cairo, Egypt
[2] OMV Upstream, Exploration, 1020 Vienna, Austria

*Correspondence to*: William Bosworth (bill.bosworth@apacheegypt.com)

**Abstract.** Folds associated with inverted extensional faults are important exploration targets in many basins across our planet. A common cause for failure to trap hydrocarbons in inversion structures is crestal breaching or erosion of top seal. The likelihood of failure increases as the intensity of inversion grows. Inversion also decreases the amount of overburden, which can adversely affect maturation of source rocks within the underlying syn-extensional stratigraphic section. However, many rift basins are multi-phase in origin, and in some cases the various syn-rift and post-rift events are separated by multiple phases of compression. When an inversion event is followed by a later phase of extension and subsidence, new top seals can be deposited and hydrocarbon maturation enhanced or reinitiated. These more complex rift histories can result in intra-basinal folds that have higher chances of success than single-phase inversion-related targets. In other basins, repeated inversion events can occur without significant intervening extension. This can also produce more complicated hydrocarbon maturation histories and trap geometries. Multiple phases of rifting and inversion affected numerous basins in North Africa and the Black Sea region and produced some structures that are now prolific hydrocarbon producing fields, and others that failed. Understanding a basin's sequence of extensional and contractional events and the resulting complex interactions is essential to formulating successful exploration strategies in these settings.

## 1 Introduction

Although the concept of structural inversion has now existed for over a century (Lamplugh, 1919), it was Glennie and Boegner (1981) who specifically used this term to describe the formation of a particular structure in the southern North Sea. Shortly thereafter Bally (1984) generalized the concept. The importance of inversion tectonics to both academic researchers and industry experts was quickly recognized as shown by several seminal works (e.g. Cooper and Williams, 1989; Buchanan and Buchanan, 1995). For a discussion of the past 30-year history of positive inversion as a structural geology concept see Kley and Krzywiec (2020).





Positive structural inversion entails removal of extensional ("normal") offset on a fault and the formation of associated anticlines. These features are of considerable interest to oil and gas explorationists. The opposite process of negative inversion in which contractional ("reverse") offset is removed is generally of less economic significance. "Positive inversion", or just simply "inversion" for the remainder of our discussion, has many effects on all aspects of petroleum systems: maturation, migration, trapping and sealing. A certain combination of these effects could either improve or degrade

the pre-drill risk profile of a hydrocarbon exploration target (e.g. Macgregor, 1995, Turner and Williams, 2004; Cooper and Warren, 2010, Bevan and Moustafa, 2012; Tari et al., 2020). Failure to recognize the effects of inversion on a basin's geologic evolution can have a disastrous impact on an exploration program.

Inversion tectonics become increasingly complex whenever there are multiple phases of extension or shortening, as compared to the one-time extension/shortening cycle. The general aim of this paper is to provide examples of this

complexity by highlighting exploration programs that targeted structures that experienced very different inversion histories. The Western Desert of northern Egypt is selected to show a case in which multiple phases of compression were separated by multiple syn- and post-rift events. In contrast, the NW Black Sea has a rift basin fabric that was formed by multiple phases of extension during the Triassic to Cretaceous but then was inverted by multiple phases of shortening during the Cenozoic without any intervening extensional periods. Besides illustrating the multi-faceted impact on the petroleum system elements,

these case studies could also potentially serve as exploration templates in basins with similar tectonostratigraphic evolution.

## 2 Western Desert

The Egyptian Western Desert includes all the land west of the Nile Delta, Nile River and Lake Nasser to the border with

Libya (Fig. 1). The first economic oil or gas discovery in the Western Desert was Alamein field, found by Phillips Petroleum Company in 1966 (Metwalli and El-Hady, 1975; Egyptian General Petroleum Corporation, 1992). The reservoir interval is Aptian age dolostone, located in an ENE-WSW trending elongate faulted anticline. Although not discussed in early interpretations of the field, Alamein is an inverted structure, with shortening imposed in the Late Cretaceous. Other inversion-related traps were subsequently found, including the first oil and gas discovery in the massive Abu Gharadig basin

in 1969 (Abu Gharadig field; El Gazzar et al., 2016). Years later, when the inverted Qarun field was found in 1994 it marked the largest Egyptian discovery in about a decade (Abd El-Aziz et al., 1998). Unfortunately, many unsuccessful wells were also drilled on the subsurface crests of other large Western Desert inversion folds. Failure was often attributed to the erosion of top seal and breaching of the underlying reservoir objectives.

Alamein, Abu Gharadig, Qarun, and most other Western Desert inverted structures were formed by extension and associated

subsidence in the Late Jurassic to Early Cretaceous, followed by shortening in the Late Cretaceous to Eocene. The Late Cretaceous inversion, or "Santonian event", was by far the most significant compressional tectonics to affect the Western Desert during the Phanerozoic, but there were other compressional events. We first briefly outline the tectonostratigraphic





history of the Western Desert, then describe less frequently observed inversion in the Early Cretaceous, followed by an example of the main Santonian inversion.

## 2. 1 Geologic setting

The Phanerozoic history of the Western Desert was shaped by the opening of first Paleotethys and then Neotethys, which morphed into the modern Mediterranean Sea when the seaway between Arabia and Eurasia closed about 15 Ma. Extensional structures related to Paleotethys are present in the subsurface of the Western Desert but are presently not well known. Neotethyan rifting, however, left a complex legacy of multi-phase basins along the northern margin of Gondwana (Fig. 1). Further west in Algeria and Tunisia initial opening began in the Permian, and by the Triassic had reached northern Egypt and the Levant and a seaway extended into Syria (Şengör, 1979; Stampfli et al., 2001; Garfunkel, 2004; Berra and Angiolini, 2014). Permian and Triassic continental strata are encountered in wells in the far Western Desert and in outcrops along the Gulf of Suez. Like the Paleozoic section, relatively little is known regarding the structural setting of these units.

The earliest well-defined rifting event in the Western Desert occurred during the mid- to late Jurassic and established the general basin configuration that persisted through most of the Mesozoic (Keeley and Wallis, 1991; Guiraud, 1998). Most faults active in the Jurassic are oriented E-W to ENE-WSW with an ~N-S extension direction (Fig. 2). This structuration is generally attributed to the distal effects of the continued opening of Neotethys further to the north. However, potential fields and seismic datasets acquired over the past several decades have been interpreted to suggest that the Eastern Mediterranean basin opened in a WNW-ESE direction and that the Egyptian margin was a transform boundary (Longacre et al. 2007). Resolving the apparent disconnect between the Egyptian offshore and Western Desert onshore basin kinematics will be important to better understanding the geodynamic evolution of NE Gondwana.

In the western Faghur and Shushan sub-basins, Jurassic rifting was marked by an early phase of volcanism, mostly in the form of local basaltic flows, tuffs and volcaniclastics (Abbas et al., 2019). The volcanics are overlain and interfinger with siliciclastic rocks that are ascribed to the Khatatba Formation (Norton, 1967; Fig. 2). The Khatatba Formation is both an important reservoir objective and the most important source rock in the Western Desert (Keeley et al., 1990).

Western Desert 'Jurassic' rifting was relatively short-lived and ended in the earliest Cretaceous, spanning a period of ~7 Myr or less (~147-140 Ma). The syn-rift stratigraphy varies dramatically in thickness and facies from sub-basin to sub-basin. In general, the section is much thinner in the west and south and thickens toward the north. At the end of the Jurassic to earliest Cretaceous, a widespread but brief marine incursion resulted in the deposition of Masajid Formation open marine limestone facies over most of the Western Desert, except on a few, high-standing platform areas (Fig. 2; Norton, 1967; Keeley et al., 1990).

Immediately following Masajid flooding, during which active extensional faulting is not recognizable in most sub-basins, a second phase of rifting initiated with strata assigned to the Alam el Bueib Member of the Burg el Arab Formation (Fig. 2; Norton, 1967). This is the most pronounced extensional phase in almost all Western Desert sub-basins and lasted about 14





Myr (~139-125 Ma). Extension was also initially N-S directed, but midway through the rift event, extension rotated to NE-
     SW (Fig.2).

In addition to the strong rotation of the extensional stress field, which is also recognized in many other basins of north and
central Gondwana (Guiraud and Bellion, 1995; Guiraud, 1998; Guiraud and Bosworth, 1999; Guiraud et al., 2001, 2005), the
Western Desert experienced a pulse of compression at about 138 Ma, which we refer to as the Late Cimmerian event (Fig.

2). This shortening only affected a small number of faults, an example of which is discussed below.

The Alam el Bueib phase of rifting, like the Khatatba, ended with a second even more regionally extensive marine flooding
event, which deposited the Alamein and Dahab Members (Norton, 1967). Extension renewed in the mid-Aptian at about 120
Ma, and marine deposition was replaced by predominantly fluvial deposits of the Kharita Member and Bahariya Formation
(Said, 1962; Norton, 1967). Kharita-Bahariya rifting was prolonged, lasting about 20 Myr (~120-100 Ma), but generally

occurred at slower extension rates that gradually dissipated in lower Bahariya times. In other parts of Gondwana, the Albian-
     Aptian was the most important phase of extension, as was the case in much of the central African rift system (Schull, 1988;
     McHargue et al., 1992).

Sea-level rise in the Cenomanian and Turonian resulted in flooding of all the Western Desert and establishment of an epeiric
sea that would last into the early Cenozoic (Said, 1962; Kerdany and Cherif, 1990). These marine strata are assigned to the

upper Bahariya and Abu Roash Formations (Fig. 2; Norton, 1967) and were deposited during a relatively quiescent period in
     the Western Desert. In the Sirt basin to the west (Fig. 1), this was a time of significant extension and subsidence in its NW-
     SE trending sub-basins (Abadi et al., 2008). The Western Desert calm was abruptly terminated at 84 Ma with the onset of the
     main pulse of regional basin inversion, the Santonian event (Moustafa and Khalil, 1995; Guiraud and Bosworth, 1997;
     Guiraud, 1998; Bevan and Moustafa, 2012). Santonian compression was of true plate-scale significance, as was recognized

long ago by Burke and Dewey (1974).

Santonian inversion can be interpreted to be a consequence of a change in relative movement between the Eurasian and
African plates, with N-S divergence switching to N-S slightly oblique convergence (Savostin et al., 1986; Le Pichon et al.,
1988). Convergence continues to the present-day and was manifest in North Africa by a series of compressional pulses,
interspersed with periods of quiescence or extension that were spatially complex (Bevan and Moustafa, 2012). The most

pronounced post-Santonian shortening occurred at the end-Cretaceous and within the late Eocene, corresponding to coeval
     compressional maxima in the Alpine belt of Eurasia (Fig. 2; Guiraud et al., 1987; Guiraud and Bosworth, 1997; Guiraud,
     1998).

During and following Santonian inversion, shallow marine carbonate environments continued across the Western Desert
with deposition of the Khoman Formation (Fig. 2; Norton, 1967). The Khoman, which is commonly a chalky facies, is

completely missing from the crests of some major Santonian inversion structures. Apollonia Formation (a term borrowed
     from Libyan stratigraphy) limestone deposition commenced following the base Cenozoic unconformity and generally
     continued until the late Eocene deformation when the northern Western Desert epeiric seas began to retreat and siliciclastic
     deposition returned (Dabaa Fm.; Norton, 1967). Mixed carbonate and siliciclastic deposition continued through the



Oligocene and Miocene (Moghra and Marmarica Fms.; Said, 1962), punctuated by a very brief period of basaltic volcanism
at 24-22 Ma that was related to Red Sea rift initiation (Fig. 2; Meneisy, 1990; Bosworth et al., 2015). Most of the Western
Desert, excluding some coastal regions, experienced gradual uplift and erosion from the late Miocene to Present-day.

## 2. 2 Faghur basin Cimmerian inversion

Faghur is the westernmost sub-basin of the Egyptian Western Desert rift system (Fig. 1). Exploration started there in the late
1950's encouraged by success to the west in the basins of Libya. However, the first commercial discovery wasn't made until
2006. The only functioning source rock in the Faghur basin is the Khatatba Formation (Bosworth et al., 2015; Fig. 2), which
was deposited during the short-lived late Jurassic first phase of rifting.

Extensional faults that affect the Khatatba and immediately overlying Masajid Formation strike predominantly in two
orientations: E-W or ENE-WSW (Fig. 3). Like the other main sub-basins of the Western Desert, most of these faults dip to
the south, which is significant as the coeval Neotethyan margin stepped down to the north. The south dip probably reflects
reactivation of a pre-existing basement or Paleozoic (Hercynian?) structural fabric.

The only places where an early phase of shortening and inversion have been observed are on a few of the ENE-WSW
striking faults, as along the Tayim-Phiops trend (Fig. 3). There the inversion affected a small segment of the fault system that
dipped NW, just north of the large Kalabsha horst block. Early syn-rift growth on this fault was small but resolvable (Fig. 4).
Inversion occurred during deposition of the basal units of the Alam el Bueib Member, so in very early Cretaceous times.
Based on this timing and in accordance with the better documented tectonic phases of SE Europe we designate the inversion
a "Late Cimmerian" event (Nikishin et al., 2001; Stampfli et al., 2001). Minor folding and local erosion of this age have been
observed elsewhere in North Africa, the Benue trough, the Levant margin and Arabian platform (summarized in Guiraud et
al., 2005).

Alam el Bueib syn-rift phase 2 strata drape over and seal the inversion anticline (Fig. 4). Differential compaction across the
structure affected most of the mid- and upper Alam el Bueib strata resulting in four-way dipping unfaulted closures higher in
the section. In detail, the hinge of the Phiops fold is doubly-plunging and not exactly parallel to the underlying contractional
fault (Fig. 5). The hinge curves away from the ENE-WSW striking fault becoming NE-SW trending suggesting that the
shortening direction was approximately NW-SE oriented. Along strike several other smaller inversion anticlines are
recognized, and to the north the fold trend steps to the east across another major down-to-the south early extensional fault.

In the Faghur basin oil migration commenced in the Late Cretaceous (Bosworth et al., 2015; Abdelbaset et al., 2019), long
after the Late Cimmerian inversion structure was already formed. Reserves are trapped in both pre- and post-inversion
siliciclastic reservoirs. The amount of shortening at Phiops is not large, although it did remove all the early extension on the
fault and all units now display reverse offset. The Phiops inversion is restricted to a single fault trend and had no noticeable
effect at the scale of the Faghur sub-basin. No reserves have so far been recovered from the over-thrust footwall block.





The products of younger inversion are present in the Faghur basin but are very minor. Structures formed by Santonian shortening include small folds of the Abu Roash strata (Fig. 2) along the large basin-bounding faults. This was of no consequence to the hydrocarbon system of the sub-basin. Slightly more significant was renewed NE-SW extension and accompanying sedimentation during the Campanian and Maastrichtian, which provided additional overburden and therefore helped to accelerate maturation of the deep Khatatba source rocks. Late Cretaceous NW-SE shortening and NE-SW

extension were probably at least in part coeval at Faghur.

**2. 3 Alamein-East Abu Gharadig basins Santonian inversion**

Late Cretaceous shortening in the Western Desert has been extensively documented, both in outcrop (Moustafa, 1988; Abdel Khalek et al., 1989; Moustafa et al., 2003) and the subsurface (Kerdany and Cherif, 1990; Moustafa et al., 1998; Yousef et al., 2010, 2019; Bevan and Moustafa, 2012). In the eastern sub-basins of the Western Desert, inversion is manifest at both

the scale of individual faults and across complete sub-basin profiles. Shortening was intense in the Alamein, Abu Gharadig, and Matruh sub-basins (Fig. 1), but as discussed above largely absent from the westernmost regions. Near the border with Libya, almost all the Late Cretaceous shortening occurred further to the north in Cyrenaica which acted as a promontory or indenter during the Eurasia-Gondwana collision (Bosworth et al., 2008).

A regional transect of the Alamein and East Abu Gharadig basins illustrates the scale and significance of Late Cretaceous

and younger inversion in the eastern Western Desert (Fig. 6; see also Bevan and Moustafa, 2012, their fig. 19.7). The stratigraphy of the eastern sub-basins is very similar to that of Faghur in its overall framework. In detail, several differences can be noted: 1) the pre-Jurassic stratigraphic section is much thinner or absent completely; 2) particularly in the north, depositional facies in the Jurassic and Cretaceous tend to display more marine affinities; 3) thickness variations in the Cenozoic section are much more dramatic, in part due to the effects of late inversion; and 4) a gentle, regional northward tilt

of the late Miocene to Holocene section, particularly in the Alamein basin (not observed in Faghur).

The most pronounced shortening at the longitude of Figure 6 occurred at the Mubarak inversion where crystalline basement now structurally overlies part of the early syn-rift stratigraphy. The inverted fault at this position was not the original basin-bounding fault, but rather cuts through the axis of the early basin. The area is covered by good quality 3D seismic reflection data, and along-strike the basin-bounding and inverted faults merge to produce a more 'typical' inverted extensional fault.

Other less prominent inversion structures, more akin to the scale of Early Cretaceous Cimmerian shortening described at Faghur, are present at Misaada and Gondul (Fig. 6).

The most prominent inversion-related unconformity across all these eastern basin structures initiated in the Santonian. Pronounced on-lap of the Campanian-Maastrichtian Khoman chalk onto the Mubarak fold is evident in seismic lines (Bevan and Moustafa, 2012). A second dramatic Mubarak unconformity developed at the end of the Mesozoic, indicating renewed

shortening and denudation that continued into the Late Eocene.





The total Jurassic to Recent stratigraphic thickness of the Alamein, East Abu Gharadig and Faghur basins are quite comparable, generally 5-6 km along the main basin axes. However, the geothermal gradients at Alamein and East Abu Gharadig are much higher than at Faghur, and therefore oil generation commenced earlier, generally in the mid-Cretaceous. Migration was well underway by the time of the Santonian inversion, and more so for the later pulses of compression.

Breaching of some reservoirs that had already trapped hydrocarbons was inevitable. Fortunately for the inversion structures in Figure 6, numerous reservoir horizons remained intact and Early Cretaceous syn-rift exploration targets were successful.

**2. 4 Significance of multiple inversion events to Western Desert hydrocarbon systems**

The exploratory wells drilled on the inversion structures of Figures 4 and 6 were all successful. Along strike, other wells were not so lucky. Other parts of the Mubarak inversion were uplifted and eroded more deeply than at the EB-32A location.

In some cases, wells encountered reservoirs with residual hydrocarbons suggesting that oil migration and trapping occurred and then was lost. The Phiops, Misaada, and Gondul trends are much smaller structures, and all display very complex local fault patterns. Offset or similar play types to these wells were not always successful either.

In addition to breaching structurally shallow reservoirs, the large Western Desert inversions such as Mubarak also interrupt hydrocarbon maturation processes, at least over the region undergoing significant uplift. Estimating how much stratigraphic

section was removed, rather than non-deposited, is an extremely complex and difficult problem to address. The potential effects on paleo-heat flow are another consideration, generally not well-constrained or even considered.

Basin-scale inversions like Mubarak also drastically impacted migration pathways (Bevan and Moustafa, 2012). Prior to Santonian inversion, almost all hydrocarbons being generated and expelled from the Jurassic Khatatba Formation in the East Abu Gharadig basin were flowing through carrier beds up-dip to the south, toward Misaada and Gondul (Fig. 6). During and

after Santonian inversion, a large part of the basin axis increasingly dipped to the south, refocusing migration to the north. Understanding these changes in migration paths, which can occur at both local and regional scales, are important to successful exploration strategies.

**3 Black Sea**

The Black Sea is classically divided into two separate basins, the Western and Eastern Black Sea basins (WBSB and EBSB),

with the divide formed by the Andrusov and Arkhangelsky ridges and the Tetyaev high (collectively, the Mid-Black Sea high) that trend approximately north–south in the central part of the Black Sea (Fig. 7). Our study area is in the broader Gulf of Odessa (or Odessa Shelf) located in the northern part of the WBSB. Our data base is composed of about 8,000 km of legacy 2D reflection seismic data and close to 90 wells drilled for hydrocarbon exploration purposes.

There are several examples of inversion structures with associated hydrocarbon fields in the Black Sea. In the Histria trough

of Romania (Fig. 7), multiple phases of Cenozoic inversion have been described (Morosanu, 2002; Dinu et al., 2005). Drilling in the Romanian Black Sea started in 1976 and led to the discovery of the Lebada field in 1981 which has a trap





with an element of inversion (Krezsek et al., 2018). In the Turkish sector, the biogenic gas field of Akcakoca was discovered by Turkish Petroleum in 1976 (Fig. 7). Subsequent drilling proved the commerciality of this gas find reservoired in Middle Eocene turbidites. The trap for this field is an inverted anticline (Robinson et al., 1996; Alaygut et al., 2004; Menlikli et al., 2009) due to the regional shortening associated with Late Eocene basin-scale inversion. In the Gulf of Odessa of Ukraine the first offshore discovery was Golitsyna in 1975 (Fig. 7), an anticline with Paleocene chalk and Oligocene sandstone reservoirs displaying renewed episodes of inversion after the largest Late Eocene one (Robinson and Kerusov, 1997; Khriachtchevskaia et al., 2009, 2010).

After briefly describing the tectonostratigraphic evolution of the WBSB, we provide a modern, depth-converted regional-scale seismic illustration of the multiple inversion periods in the Karkinit basin, Shtoromoe graben and Kalamit high area (Fig. 7). An additional legacy 2D seismic example was selected to show the untested deep gas potential along the northern perimeter of the inverted Karkinit basin. Finally, we highlight the un(der)explored intra-Maykop stratigraphic play potential. This is directly linked to the strongest Late Eocene inversion episode in the Black Sea area which created pronounced accommodation space differences.

## 3.1 Geologic setting

The Black Sea is a Cretaceous basin complex superposed on the northern margin of the Tethys/southern margin of Laurussia (Nikishin et al., 2001; Okay and Nikishin, 2015). The Mesozoic pre-rift tectonostratigraphy of the WBSB is quite complex as it has elements of Early to Middle Triassic rifting, Late Triassic Early Cimmerian orogenesis, Jurassic back-arc extension, and the Late Jurassic Late Cimmerian regional compressional phase (Fig. 2; Nikishin et al., 2001). These alternating extensional and compressional cycles produced inverted structures, like those of the Triassic rifts on the Scythian Platform and in Dobrogea (Saintot et al., 2006), but these are typically poorly understood subsurface features.

The Black Sea basin complex is traditionally thought to be a marginal or back-arc basin with active rifting beginning in the mid-Cretaceous (Finetti et al., 1988; Nikishin et al. 2015a,b). In terms of geodynamic models of modern back-arc basin formation, this extension was driven by slab roll-back (Stephenson and Schellart, 2010). However, a debate in the literature is still ongoing regarding not only the geodynamic reason for the basin opening, but also its timing and kinematics (Tari, 2015).

The Black Sea basin complex opened in a complex manner and Tari (2015) distinguished two major rifting periods within the Cretaceous. Intitial rifting started as soon as the Barremian and became regionally widespread in a "wide-rift" mode by the Aptian-Albian (syn-rift stage 1; Fig. 2) with numerous rift sub-basins trending NW-SE or E-W (Robinson and Kerusov, 1997; Krezsek et al., 2018). There is surface and subsurface evidence for Albian volcanics in the area, including western Crimea (Nikishin et al., 2013) and the mostly andesitic volcanism appears to be limited to the E-W trending Karkinit basin. The trend of rifting changed to NE-SW at the end of the Albian and a new rifting period occurred during the Cenomanian to Santonian (syn-rift stage 2; Fig. 2). During this time a "narrow-rift" style of much larger scale, regional volcanic back-arc





extension was superimposed on the Early Cretaceous, mostly non-volcanic extensional system. By the Santonian, the WBSB

opened to its full extent and the top Santonian is considered as the break-up unconformity in our study area (Khriachtchevskaia et al., 2010).

Since the first basin-wide distributed volcanics are Turonian in age, the WBSB evolved as a *sensu stricto* back-arc basin only during the Turonian-Santonian interval. The subsequent widespread Campanian volcanism in the Pontides, and its assumed equivalent in the Turkish offshore (Nikishin et al., 2015a), was interpreted by Tari (2015) as being arc-related, post-dating

the opening of the WBSB.

The Uppermost Cretaceous and Lower Paleogene (Paleocene to Middle Eocene) stratigraphy of the Odessa Shelf is dominated by chalks (Figs. 2,8), reflecting tectonic quiescence in a post-rift setting. The first compressional event disrupting the waning subsidence pattern happened at the end of the Middle Eocene at about 38.6 Ma ((Khriachtchevskaia et al., 2010) and the deposition of carbonates was replaced by shales (Figs. 2,8). During the Late Eocene at about 35.4 Ma, another basin-

wide shortening episode produced the bulk of the inverted structures (Khriachtchevskaia et al., 2010). This "Pyrenean" event (Fig. 2) is considered as the most significant one in the broader Black Sea area and it can be correlated with the last phase of overthrusting in the Balkans (Doglioni et al., 1996; Bergerat et al., 2010). The Crimean Mountains also experienced shortening-related uplift during this time based on apatite fission-track studies (Panek et al., 2009).

Regionally, the Oligocene to Lower Miocene Maykop Formation (Figs. 2,8) postdates the two Eocene discrete inversion

events as can be deduced from the onlap geometries seen on reflection seismic data (Fig. 9). The early and Middle Miocene saw another two inversion events (circa 16.3 and 10.4 Ma) in our study area (Khriachtchevskaia et al., 2010). The pronounced diapir-looking structure (Gamburtsev) in the middle of the regional seismic line (Fig. 9) is an extreme example of the multiple contractional reactivation of an already existing inverted structure. The inversion process was quite selective spatially and temporarily across the Odessa Shelf, as not all the pre-existing Cretaceous master faults were reactivated in any

given cross-section (Fig. 9). However, the large border fault on the northern margin of the Karkinit basin did experience reactivation along strike to the east in the area of the Golytsina gas-condensate field (Fig. 10). The seismic expression of both the footwall and hanging wall is clear and even the position of the null-point can be determined with confidence. The inversion clearly post-dated the Maykop Formation and therefore is post-early Miocene in age.

The Sudak folded belt offshore Crimea (Tari and Simmons, 2018) formed during the Miocene (Stovba et al., 2009; Sheremet

et al., 2016a,b) in multiple stages (Fig. 2). The corresponding Miocene compressional episodes with slightly rotating, but generally N-S oriented compressional stress fields, were documented by micro-tectonic studies in onshore Crimea (Murovskaya et al., 2014, Hippolyte et al., 2018). The challenge onshore, just like in the offshore, is that these stages or events cannot be precisely dated, i.e. with resolution less than 1-2 Myr, and separated thus far. This limitation is primarily due to the lack of Miocene sediments in the Crimean Mountains onshore and the lack of sufficiently dense sampling of the

stratigraphy in offshore industry wells.

Khriachtchevskaia et al. (2010) argued that the period of discrete inversion ended by the Late Miocene, or at least was suspended. This is contrary to the models of Robinson et al. (1995) and Nikishin et al. (2003) who suggested an accelerated





period of subsidence in the Black Sea basin complex since the Late Miocene or Pliocene, respectively, as the result of an overall N-S directed compressional stress regime down-bending the basin center. Whereas this subsidence is difficult to document given the resolution of the biostratigraphic dating, a closer look at the available seismic reflection data does provide definitive evidence for ongoing post-Late Miocene compression in the Gulf of Odessa.

We chose a legacy 2D seismic line across the Shtormovoe inversion anticline to show how this particular feature displays signs of repeated and also neotectonic shortening (Fig. 11). This broad, 20 km wide structure is a composite one at depth, i.e. below about 1 second two-way travel time it splits into two inversion anticlines, 4 and 12 km across. These correspond to earlier Eocene inversion events. With the continuous thickening of the sedimentary cover, the earlier Eocene inversion anticlines were incorporated into a broader, single Miocene to Pliocene anticline.

A key observation regards the geometry of a prograding shelf margin sequence over the apex of the structure which postdates the pronounced regional intra-Sarmatian (Khersonian) unconformity dated as circa 7.5 Ma in the Black Sea (Fig. 11; Popov et al., 2010). The clinoforms in this prograding unit are slightly back-rotated to the north and their top-laps, which should be sub-horizontal, show a c. 2-4° northward tilt (Fig. 11). There are also two onlapping reflectors on the northern flank above the prograding sequence. Given the dimensions of the structure and the timing, this back-rotation cannot be attributed to differential compaction. These observations underline the reactivation of the inversion process during the Pliocene.

The present-day stress field in the area, based on earthquake focal mechanism studies, is a compressional to strike-slip one (Murovskaya et al., 2018) which is consistent with other regional observations of ongoing N-S directed compression in the broader Black Sea area (Tsereteli et al., 2016).

## 3.2 Implications for NW Black Sea exploration

There are several hydrocarbon fields in the Gulf of Odessa and the adjacent Crimean Peninsula (Fig. 12). The Odessa Shelf was explored for the last five decades and eight gas and gas-condensate fields have been discovered, all drilled in jack-up water depth (less than 100 m). Exploration was historically focused on the inverted structural highs. The productive horizons are related to Upper Cretaceous (Maastrichtian), Paleocene, Eocene, Oligocene-Lower Miocene reservoirs found at depths of 480–3000 m (Khriachtchevskaia et al., 2009; Stovba et al. 2009). The two largest gas and condensate finds, Golitsyno, Shtormovoe, with recoverable gas/condensate reserves of 420 bcf/3 mmboe and 777 bcf/21 mmboe, respectively, are developed.

Nedosekova et al. (2008) reviewed the drilled structures and concluded that all the prospects and leads associated with simple 4-way closures have been tested. As a general observation, there seems to be a trap-fill issue as the inverted structures could hold much larger hydrocarbon volumes than the discovered resources. The map-view 4-way closures of the Golitsyno and Shtormovoe anticlines are 680 and 440 km$^2$, respectively (Sergey Stovba, personal communication, 2010) with multi-tcf recoverable gas potential. However, these structures are clearly not filled-to-spill and the observed gas columns are in the





tens of meters range. The underfilled trap issue can be explained by charge limitations, trap timing versus charge and/or by trap failure/breaching. Given the multiple, up to four inversional events shaping these anticlines individually, losses to several remigration periods between the inversions should have played a role. These risks also explain why some of the large inverted structures in the area turned out to be dry, like the prominent Gamburtsev anticline (Fig. 9).

There are two schools of thought as to finding more hydrocarbons in this seemingly mature petroleum province. One
suggestion was made by Burchell (2008) who described a new gas play type associated with the deeper part of the Golitsyno anticline, beneath the producing Lower Paleocene chalks (Fig. 10; Robinson and Kerusov, 1997). Four possible gas-charged Lower Cretaceous sand targets were considered within the rift basin fill of the Karkinit trough at 4,500 to 5,500 m depth (Fig. 10). These 3-way structural targets have a large map-view extent, on the order of about 300 km$^2$. Hydrocarbon charge was deemed to be relatively low risk by Burchell (2008) given the gas-condensate finds in adjacent fields and assuming thick
gas-mature Lower Cretaceous shales in between the sand units in the so far undrilled rift basin center. The obvious exploration risks of this deep, inversion-related play include side-seal against crystalline basement across the large inverted fault, the presence and quality of the Lower Cretaceous reservoir objectives and trap definition due to the lack of 3D seismic data. This deep play remains untested to date.

The other line of thought is represented by Nedosekova et al. (2008) emphasizing the underexplored nature of stratigraphic
and combination traps in the region, such as pinchouts along the flank of paleo-highs. To show the impact of the Eocene inversion events shaping the paleo-relief of the basin, we reproduce here the isopach of the Oligocene to Lower Miocene Maykop sequence (Fig. 12; Gozhik et al., 2010). Contrary to what basin-scale well correlations can indicate, incorporating data points from basin highs (Fig. 8) the Maykop isopach shows dramatic variations between 0 and 1700 m across the Gulf of Odessa (Fig. 12). Given this range, we interpret a deepwater sedimentary environment for the Karkinit trough.
Whereas the Maykop sequence overall is dominated by shales (Fig. 8), as in the rest of the Black Sea (Tari and Simmons, 2018), there are reservoir quality deepwater sandstones in it, as in the Krymska field (Fig. 12) and in the undeveloped Subbotina oil discovery south of the Kerch Peninsula (Fig. 7), reported by Khriachtchevskaia et al. (2009) and Stovba et al. (2009), respectively. Therefore, we speculate that future regional-scale 3D seismic surveys could image potential longitudinal and transversal intra-Maykop turbiditic systems within the Karkinit trough offering various stratigraphic traps
along the basin margin (Fig. 12).

## 4 Discussion

Interesting similarities exist between the tectonostratigraphic evolution of the Western Desert and the Western Black Sea (Fig. 2) even if these two areas located some 2000 km apart. Whereas the relative chronology of alternating extensional and compressional periods differs in many respects, several of the distinct inversion events appear to be the same. In particular,
the earliest Cretaceous "Late Cimmerian" and late Eocene "Pyrenean" phases correspond to the same intra-plate shortening episodes. Inversion therefore occurred synchronously over many adjacent lithospheric plates. This indicates that horizontal



stress transmission occurred through well-coupled plate boundaries, in our case between the African-Arabian, Anatolian and Eurasian plates. The question then becomes how far can a certain peak in the "inter-plate" horizontal stress reactivate pre-existing extensional fabric and cause detectable structural inversion?

Intuitively, when most or all the plates were in close contact with each other in large continental plate collages (i.e. Gondwana, Pangea) the same intra-plate stress signal could have been transmitted across entire continents and had a "global" impact. This would support the early perception of Stille (1924) who assumed the existence of global orogenic phases. He based his observations mostly on data from Europe and North Africa which could be the expression of intra/inter-plate stress peaks transmitted across this region throughout most of the Phanerozoic. More recently Guiraud (1998) and others (Guiraud

et al., 1987; 1992; 2001, 2015; Guiraud and Bellion, 1995; Guiraud and Bosworth, 1997) have similarly documented how precisely both Phanerozoic extension and shortening/inversion events can be correlated across Gondwana and into nearby continental plates.

Another open-ended question relates to the duration of these events. Are these phases, periods or discrete events? If the horizontal stress peaks are caused by sudden plate movement changes, are they geologically instantaneous, i.e. on the order

of 10-100 kyr, or more transient in nature, i.e. on the order of 100 kyr to 1 Myr? The duration and the rate of deformation during these inversion events have direct impact on some of the petroleum system elements of any inverted structure. A well-constrained understanding of their temporal extent would be very beneficial in any given basin analysis.

However, the Black Sea inversion structures do differ from those of the Western Desert in several important ways. In the Western Desert, the Late Cimmerian and Santonian inversions were separated by several phases of very significant

extension-driven subsidence. Inversion in the NW Black Sea was more rapid fire, quickly superimposed compressional episodes. Also, the ratio of the post-rift (up to the stratigraphic level of the latest significant inversion event) versus syn-rift basin fill is much greater in the Black Sea than in the Western Desert. Consequently, the latest Pliocene to neotectonic inversion in the NW Black Sea produced buckle folding of the thick, post-rift sedimentary cover instead of the "classic" reverse-fault bounded "Sunda-folds" (Eubank and Makki, 1981) that are more typically observed when the post-rift sequence

is still relatively thin at the time of inversion. Earlier Black Sea inversion anticlines with shorter wavelength were gradually incorporated into longer wavelength folds as the result of the thickening sedimentary cover and the repeated inversional periods. The multiple Black Sea hydrocarbon remigration episodes from older traps to relatively recently formed ones appears to be the main reason for the underfilled or dry nature of most structures in the NW Black Sea basin. Breached and leaky inversion traps are similarly a cause of failure in the Western Desert, but the abundance of pre-inversion seal-reservoir

pairs has resulted in a higher exploration success rate.

**5 Conclusions**

We have presented an example of a petroliferous basin in which multiple tectonic shortening/inversion events were separated both in time and stratigraphic position by major rift events – the Western Desert – and one where multiple phases of



inversion were superimposed on older, pre-existing rift sequences – the NW Black Sea. Other tectonostratigraphic sequences
can be imagined, and no two real basins will be identical. Despite the great range of variations that may exist, some general
conclusions can be drawn and depicted schematically (Fig. 13). These include:

1) Shortening/inversion events that occur early in a basin's extensional history are likely to produce viable traps for
hydrocarbons, in pre-, syn-, and immediate post-inversion (draping) strata because although some reservoirs may be
breached, the structure will be covered and healed by later syn-rift fill. Furthermore, early in the basin history
hydrocarbons will not yet generally have started to migrate, so overall loses from the system are minimized.

2) Early inversion events can delay hydrocarbon maturation of underlying pre- or early syn-rift source rocks due to
denudation of strata, but only if the inversion is basin-scale. This is unlikely to be significant if shortening is mild
and reverse movement is restricted to small-offset faults.

3) Inversion, whether early or late, can dramatically impact migration pathways emanating from pre-inversion source
rocks. This can occur at the scale of individual fault blocks or entire basins (see further discussion in Bevan and
Moustafa, 2012). Given that most extensional basins take the shape of large-scale half grabens, pre-inversion
migration will generally be from basin axes up-dip toward the flexural margin. Inversion can re-direct migration
toward the faulted margin and fill previously unsourced structures.

4) Late shortening/inversion events will generally have more severe impact on top seal integrity because there is less
chance for post-inversion deposition of new top seals. This risk is accentuated when there are multiple,
superimposed late inversion cycles that repeatedly disrupt reservoir stability.

**Data availability**

Some of the seismic lines used in this study are confidential and not available publicly.


**Author contribution**

William Bosworth and Gabor Tari wrote the text, prepared the figures, and compiled the manuscript.

**Competing interests**

The authors declare that they have no conflict of interest.

**Acknowledgements**

We thank Jonas Kley and Piotr Krzywiec for inviting us to contribute this paper to the special issue of Solid Earth and for
their editorial efforts. Discussions concerning inversion tectonics with Albert Bally, René Guiraud, Ahmed El-Hawat, Daniel
Helgeson, Oxana Khriachtchevskaia, Andrew Robinson, Daniel Stockli, and Sergiy Stovba are greatly appreciated. The
PSDM seismic section in the Gulf of Odessa is courtesy of ION-GXT and it is gratefully acknowledged. We thank Apache
Egypt Companies and OMV for permission to publish this paper.



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



**Figure 1: Location of Egypt's Western Desert and the regional geologic setting. Box shows the location of the Faghur basin study area and the position of the regional cross-section is indicated. Increasingly milder inversion is observed moving south in the Gulf of Suez region. Similar trends are probably present in the Western Desert, but exposures of suitably aged rocks are generally lacking. A = Alamein basin; AG = Abu Gharadig basin; M = Matruh basin; S = Shushan basin. Triassic opening direction and Neotethyan oceanic/continental crustal boundary after Longacre et al., 2007. Plate boundaries (bold lines), basins and regions of inversion from Bosworth et al., 2008 and references therein.**



**Figure 2: Mesozoic to Cenozoic tectonostratigraphy of the Western Desert and Black Sea regions. In the stress field interpretations**
**only the most significant principal stress is shown, and compressional and tensional events are separated for clarity. In some basins both extensional and contractional structures are observed to have developed simultaneously. Time scale is from Ogg et al., 2016. Gr = group; Fm = formation; Mb = member; St = suite.**






**Figure 3: Inverted structure trend of the Phiops field shown on a time structure map of the eastern North Faghur basin. Mapped horizon is top Khatatba Formation. Location of the map is shown in Fig. 1. Figure 4 seismic line of section is shown by the red line. The reverse fault that inverts the structure becomes a blind fault along-strike but can be observed at deeper horizons.**





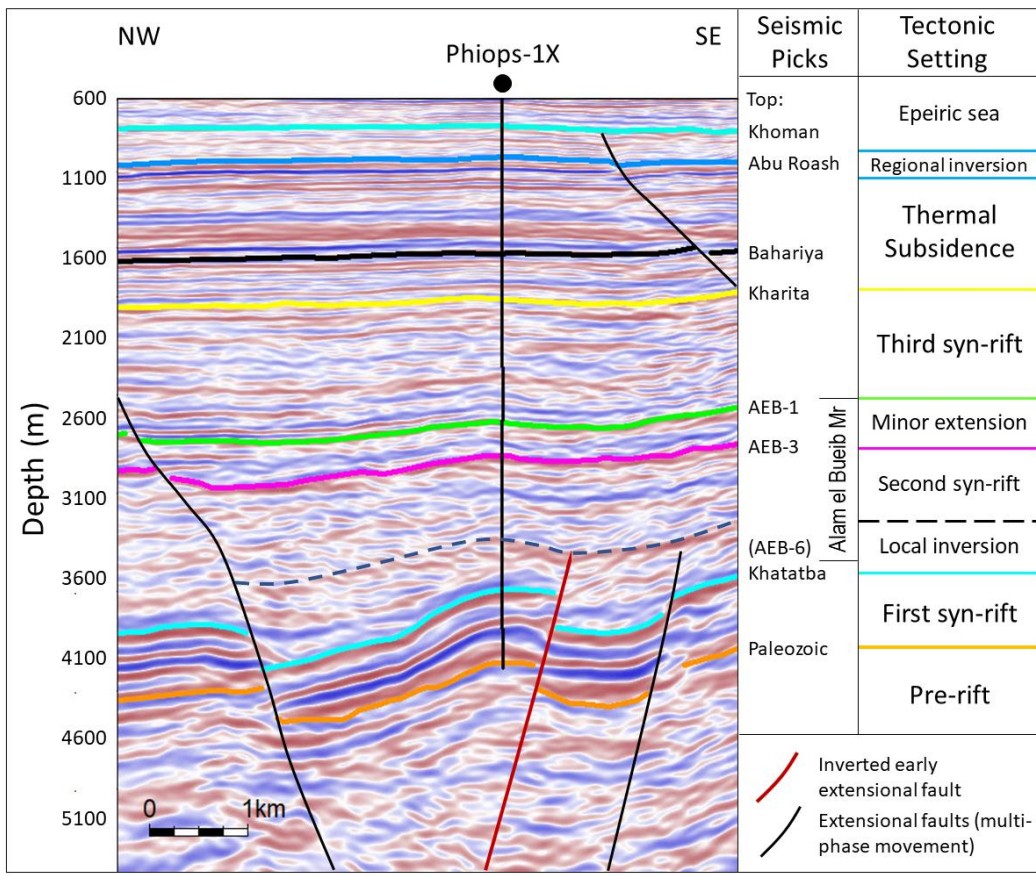


**Figure 4: Depth-migrated seismic line through the inversion structure at Phiops field. Inversion occurred during the deposition of the lower part of the Alam el Bueib Member. This was followed by differential compaction over the structure but no further shortening. The later Santonian "regional inversion" did not significantly impact this part of the Western Desert but its effects are**

**locally observable. Location is shown in Fig. 3. For ages of seismic markers see Fig. 2. ~1.7 vertical exaggeration.**


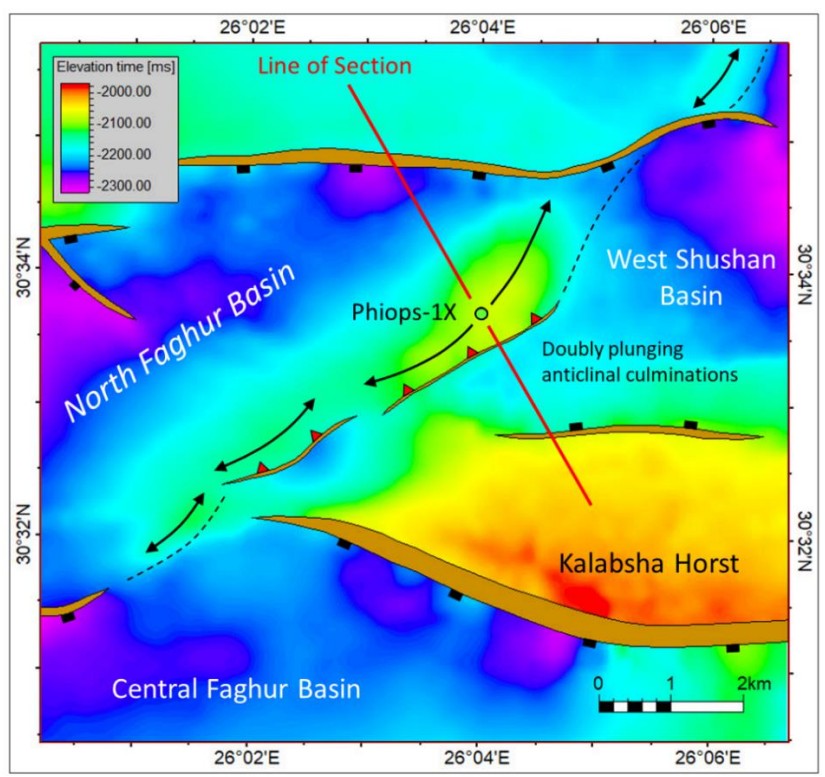

**Figure 5: Detailed time structure map of the Phiops inversion trend. Other wells have been removed for clarity. See Fig. 3 for location and legend.**


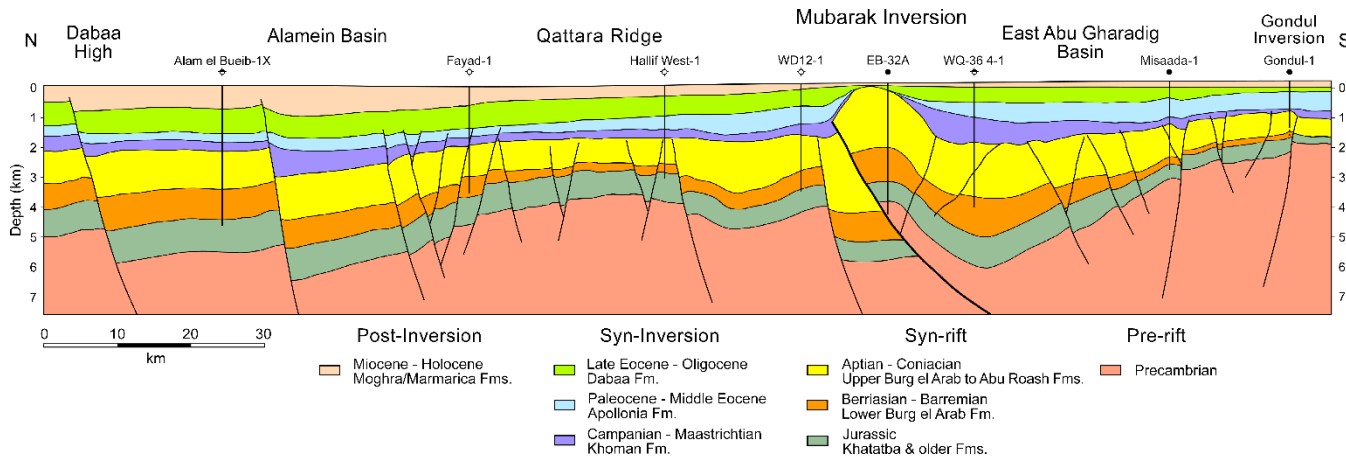

**Figure 6: Regional geoseismic section across the eastern part of the Western Desert. Location is shown in Fig. 1. The massive Mubarak inversion is one of the best examples of Western Desert Santonian inversion followed by younger pulses of shortening. After Bosworth et al., 2008. 4x vertical exaggeration.**




Figure 7: Simplified structural map of the Black Sea modified from Tari and Simmons (2018). Within the Black Sea itself, the depth to break-up unconformity is shown, adapted from Robinson (1997). White triangles represent offshore mud volcanoes and red dots represent Cretaceous paleo-volcanoes (Nikishin et al., 2015a). The locations of a regional well-correlation transect (Fig. 8), a depth-converted regional seismic line (Fig. 9), two vintage seismic profiles (Figs. 10 and 11) and a Maykop (Oligocene to Lower Miocene) isopach map (Fig. 12) are shown by red lines.







**Figure 8: Regional correlation of wells drilled on the Odessa Shelf, compiled from various sources (e.g. Gozhik et al., 2006, 2010). For location see Fig. 7. Note that none of the deep basins (e.g. Karkinit basin) have been penetrated to their full depth, unlike the basement highs (e.g. Kalamit high).**



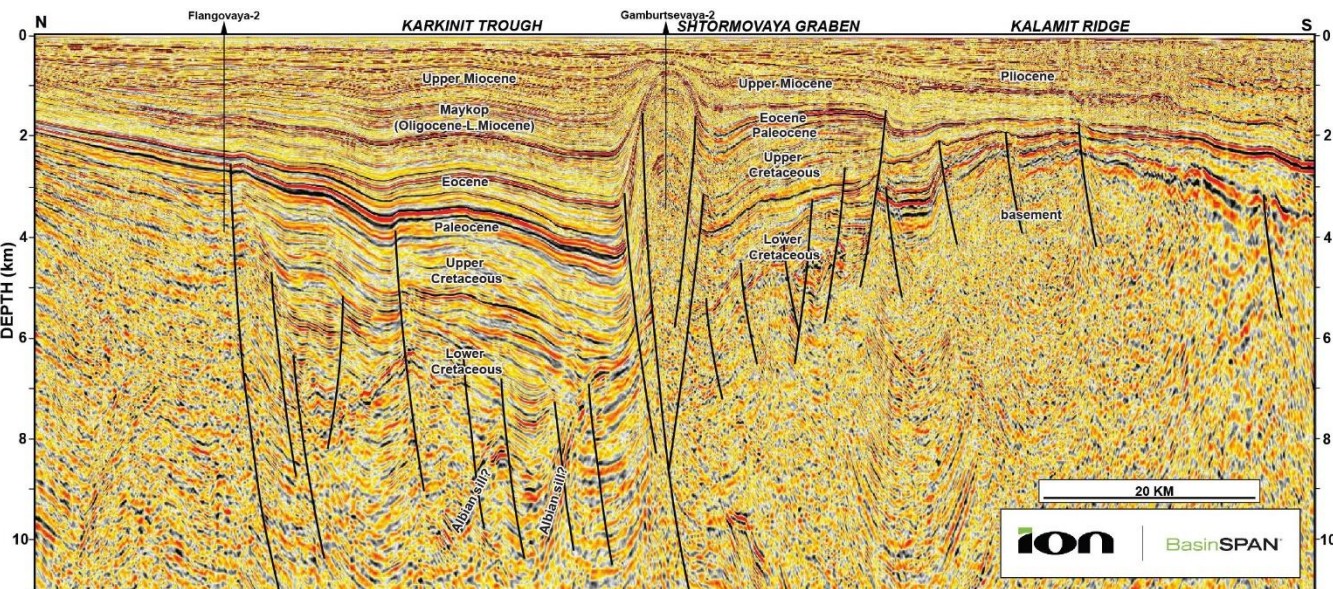

**Figure 9: Regional-scale pre-stack depth migrated (PSDM) seismic reflection profile across the Odessa Shelf, courtesy of ION-GXT. For location see Fig. 7. Note the >7500 m deep Karkinit trough in the middle of the section and the Kalamit high to the south of it. ~5x vertical exaggeration.**

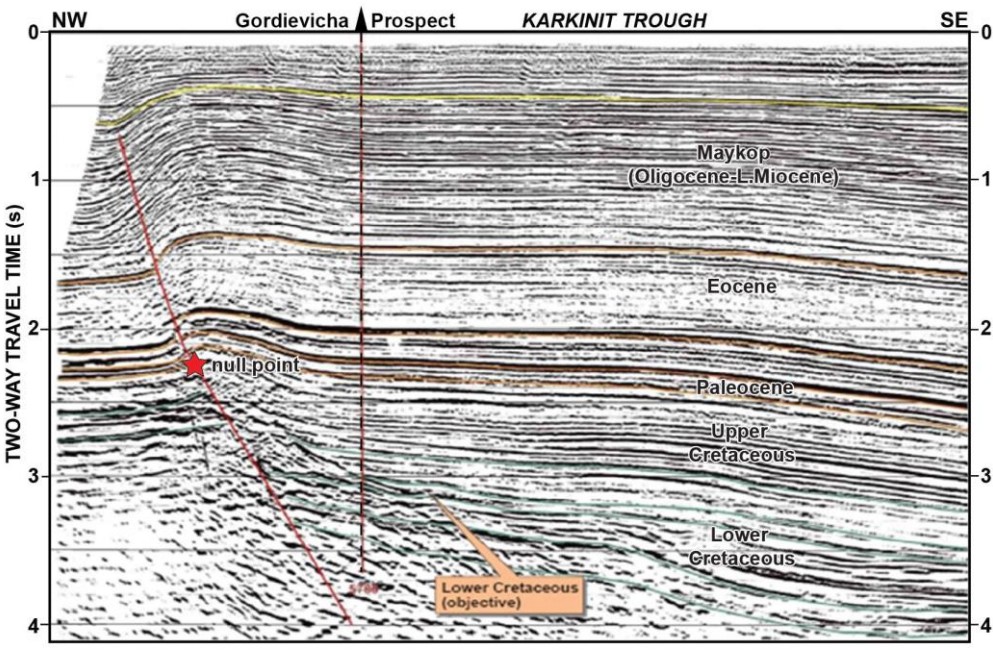

**Figure 10. Legacy 2D seismic reflection profile across the undrilled Gordievicha prospect, adapted from Burchell (2008). For location see Fig. 7. The position of the null-point is shown, *sensu* Williams et al. (1989).**





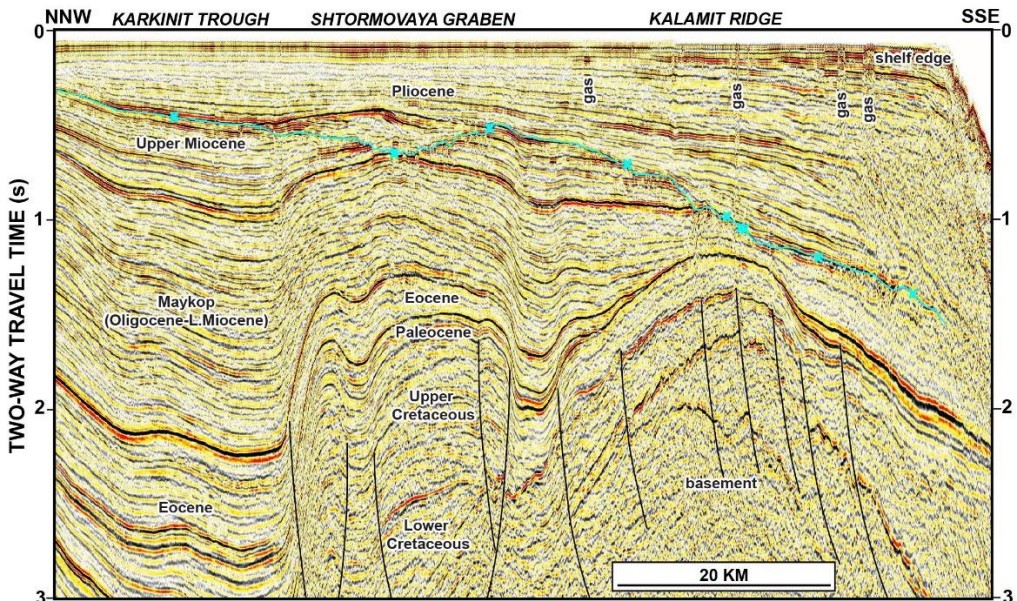

**Figure 11: Seismic reflection evidence for post-Sarmatian inversion. For location see Fig. 7. The southward prograding Pliocene sequence above the Sarmatian (Late Miocene) unconformity (shown in blue) is clearly back-rotated. This is due to the multiple episodes of inversion forming the overall structure containing the relatively small Shtormovaya field on its northern flank (e.g. Khriachtchevskaia et al., 2009). Note the gradual incorporation of the earlier Eocene folds into a much larger Miocene to Pliocene inversion anticline. Vertical exaggeration is ~6x assuming an average seismic velocity of 4 km/s.**

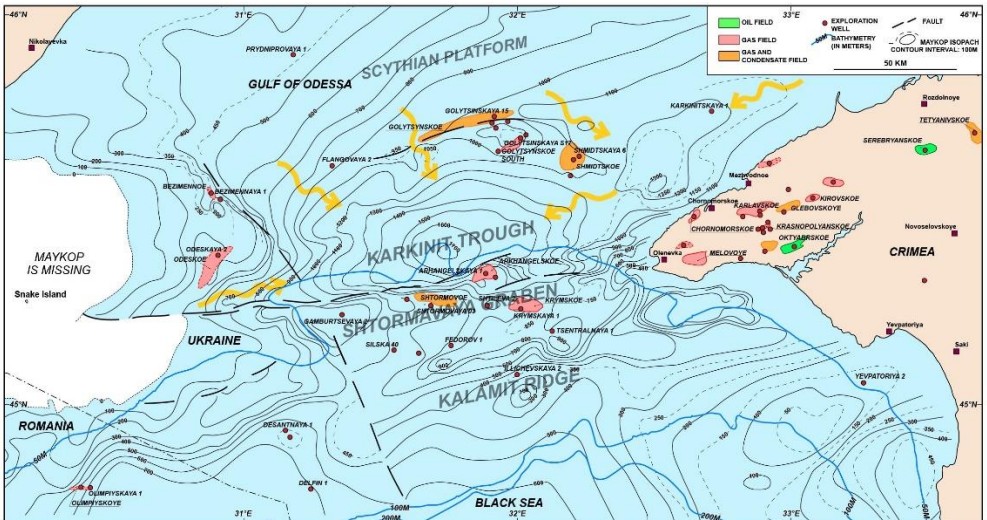

**Figure 12. Isopach of the post-inversion Oligocene to Lower Miocene Maykop Suite in the Odessa Shelf, modified from Gozhik (2010). For location see Fig. 7. Contour intervals are in meters. Note that the thickest Maykop is not captured by the currently available well control (cf. Fig. 8). The depicted sediment entry points and the deep-water distribution patterns are entirely speculative and are shown here to highlight the stratigraphic trapping potential in the Karkinit basin.**



**Figure 13. Schematic illustration of some of the possible effects of superimposed multiple phases of extension and inversion on a rift basin. For simplicity the two inversion events are shown affecting different faults, which is commonly observed in the Western Desert but will not always be the case. The various inversion phases may not be separated by significant rift cycles, as is the case in the NW Black Sea.**