# Peer review of "Hydrocarbon accumulation in basins with multiple phases of extension and inversion: examples from the Western Desert (Egypt) and the Western Black Sea"

_Solid Earth, 2020_

## Referee Comment (RC1) · Dubravko Lucic (Referee) · 28 Jul 2020

I had the privilege to review the article " Hydrocarbon accumulation in basins with multiple phases of extension and inversion: examples from the Western Desert (Egypt) and the Western Black Sea" Prepared by William Bosworth and Gábor Tari Discussion about inversion and its effects is very interesting, and it has a practical importance and huge impact in HC Exploration. Combining two different, geologically complex basins over the big distance authors highlighted the correlative events especially related to inversion, and similarities rather than differences during their geological history and evo-

lution. Looking for analogs was always not a simple task, while someone really needs deep regional geological knowledge. On the given examples from Western Desert in Egypt and Western Black Sea the wider G&G community can only benefit from. Budapest, 28th July 2020. Dr. Dubravko Lucic MOL Plc, Head of Regional Studies and New Ventures, Group Exploration

———————————————————

---

## Referee Comment (RC2) · Csaba Krézsek (Referee) · 9 Aug 2020

Dear Editor/Authors

This is a well written paper by Bosworth and Tari features the inversion history of the Western Desert of Egypt and the NW part of the Black Sea. The authors line up several examples to highlight the impact of inversion on the petroleum system and the exploration pitfalls that may occur due to poor understanding of the timing and magnitude of the tectonic events and associated sedimentary facies. Then, the similarities - and

differences - in structural style between the two studies are used to draw conclusions at global scale, useful to anyone interested exploring in inverted extensional systems. I recommend publishing the paper with minor modifications as outlined below. Best regards Csaba

Lines 50-55. Can you please specify the size of the discovered fields? Line 222 and others. Change reference Krezsek et al (2018) to Krezsek et al (2017) throughout the paper Line 255. The exact position of the break-up unconformity at basin scale is still debated being younger on the Romanian margin. Any comments to this? Line 330. Are there any clear geochemical evidences for the Lower Cretaceous SR? Alternatively, could the deeper, Tauric series be the origin of the gas/condensate?

———————————————————

---

## Author Comment (AC1) · 28 Aug 2020

We thank Dr. Lucic for his supportive remarks concerning our manuscript and his succinct summary of the objectives of our work.
* * *

---

## Author Comment (AC2) · 28 Aug 2020

Dr. Krezsek has raised several interesting questions regarding our manuscript and this has helped us to clarify some important points. We thank him for his insights and general support of our work. Concerning specific suggestions:

Lines 50-55. Can you please specify the size of the discovered fields?

- We have added in-place resources for each of the three fields discussed. This will be useful for readers to ascertain the commercial scale of the targets we are describing.

[Figure]

Line 222 and others. Change reference Krezsek et al (2018) to Krezsek et al (2017) throughout the paper

- Thank you for noting this and corrections have been made. The full reference is: Krezsek, C., Bercea, R. I., Tari, G., and Ionescu, G.: Cretaceous sedimentation along the Romanian margin of the Black Sea: inferences from onshore to offshore correlations, in: Petroleum Geology of the Black Sea, edited by: Simmons, M.D., Tari, G.C. and Okay, A.I., Geol. Soc. London Spec. Publ., 464, 211-245, 2017.

Line 255. The exact position of the break-up unconformity at basin scale is still debated being younger on the Romanian margin. Any comments to this?

- Our position, similar to the opinion expressed in the past by the reviewer himself, is stated between Lines 247-253. We slightly reworded and added the word "ultimate" on Line 255 to explain the context of the reference to Khriachtchevskaia et al. (2010): "By the Santonian, the WBSB opened to its full extent and in our study area the top Santonian is considered by Khriachtchevskaia et al. (2010) as the ultimate break-up unconformity."

Line 330. Are there any clear geochemical evidences for the Lower Cretaceous SR?

- There is evidence for Cretaceous source rocks, as Chevron and NaftaGaz jointly studied Upper Cretaceous source rocks on the Tarkhankut Peninsula with shale gas in mind in the early 2010s. We added a reference which is the only modern public domain work in this regard: Kitchka, A., Ishchenko, and Bashirov, G.: Cenomanian-Turonian calcareous black shales of the Tarkhankut Peninsula as a potential unconventional hydrocarbon shale gas play, AAPG Europe Regional Conference, Bucharest, May 19-20, 2016.

Alternatively, could the deeper, Tauric series be the origin of the gas/condensate?

- Given the great depth of the Tauric sequence beneath the study region, in the areas where it might have been preserved (see Fig. 9), we do not believe that this is a

possibility.

---

## Author Response (AR2)

**Authors' Response Manuscript se-2020-105**

**Response to review by Dr. Jonas Kley:**

Dr. Kley's comments and suggestions have been very helpful and resulted in numerous improvements to our paper. We have addressed each of the notes in his PDF file and these are shown with yellow highlighting throughout our revised text. He also raised several questions in his cover letter which we reply to in more detail here.

Tectonic overview map – we have added a map that puts the Western Desert and Black Sea into their present-day tectonic positions. This should be helpful to many readers and provides a means for better comparison of the two study areas.

Western Desert bias in concluding/summary Figure 13, now Figure 14 – we have now presented this figure as two different, rift history examples, one akin to the Western Desert and one more like the NW Black Sea. We agree this was a significant shortcoming of the original manuscript.

Structural terminology – concepts for stress and kinematics: Dr. Kley is absolutely correct that we sometimes mix what are strictly speaking meant to be stress, strain, or displacement terms. In most cases, we really should stick to strain and displacement/kinematics (a good example is the old Figure 2, now Figure 3). We have modified this throughout the text and figures.

Stille's tectonic events – has been addressed specifically using the Late Cimmerian event as an example and quoting Dr. Kley's view in the text.

Industry terms and abbreviations – we have added explanations as requested.

**Marked-up manuscript: reviewer suggested changes highlighted in yellow**

[revised manuscript text omitted]

**Rift Phase**

**Notes:**

Multiple rifting phases with source rocks deposited early in fill history

However, assumed predecessor Triassic rift basin may be too deep to generate hydrocarbons

*NOT TO SCALE!*

**Inversion Phase 1**

breach ?

First phase of inversion assumed to affect most of the major faults

Oil migration commences but a few of the traps are breached

**Inversion Phase 2**

breach ?

Second phase of inversion was selective and some of the major faults were abandoned

Asymmetric reactivation of major bounding faults caused re-migration of HCs and possibly breaching

**Inversion Phase 3**

Third phase of inversion assumed to affect most faults but without fault offsets at higher stratigraphic levels where buckle folding occurred

Deeper folds coalesced to larger folds closer to the surface and more re-migration

**Legend:**

| | |
|---|---|
| c | |
| b | Syn-rift |
| a | |
| Pre-rift | |

Syn-inversion 3
Syn-inversion 2
Syn-inversion 1

Oil & gas migration

Fault Movement
Extension  Contraction

**Figure 14b. Schematic illustration of a basin in which early extension is followed by multiple phases of inversion. The transect is**

**largely based on observations made in the NW Black Sea which experienced at least 4 distinct inversion episodes.**

---

## Author Response (AR3)

Response to requested changes by Topical Editor Piotr Krzywiec:

All available DOI's have been added to the references.

Requested changes have been made to the figures and figure captions (highlighted in yellow). Thank you for pointing these out to us.

[revised manuscript text omitted]

**Figure 2: Egypt's Western Desert and its regional geologic setting. Location is shown in Fig. 1. Box shows the location of Fig. 4 and the position of Figure 7 is indicated. Increasingly milder inversion is observed moving south in the Gulf of Suez region. Similar trends are probably present in the Western Desert, but exposures of suitably aged rocks are generally lacking. A = Alamein basin; AG = Abu Gharadig basin; M = Matruh basin; S = Shushan basin. Triassic opening direction and Neotethyan oceanic/continental**

**crustal boundary after Longacre et al. (2007). Plate boundaries (bold lines), basins and regions of inversion from Bosworth et al. (2008) and references therein.**

[Figure]

Figure 3: Mesozoic to Cenozoic tectonostratigraphy of the Western Desert and Black Sea regions. Extensional and shortening events are separated for clarity. In some basins both extensional and contractional inversion structures are observed to have developed simultaneously. Time scale is from Ogg et al. (2016). Gr = group; Fm = formation; Mb = member; St = suite.

[Figure]

**Figure 4: Inverted structure trend of the Phiops field shown on a time structure map of the eastern North Faghur basin. Mapped horizon is top Khatatba Formation. Location of the map is shown in Fig. 2. Position of Fig. 5 is indicated. The reverse fault that inverts the structure becomes a blind fault along-strike but can be observed at deeper horizons.**

[Figure]

**Figure 5: Depth-migrated seismic line through the inversion structure at Phiops field. Inversion occurred during the deposition of the lower part of the Alam el Bueib Member. This was followed by differential compaction over the structure but no further shortening. The later Santonian "regional inversion" did not significantly impact this part of the Western Desert but its effects are locally observable. Location is shown in Fig. 4. For ages of seismic markers see Fig. 3. ~1.7 vertical exaggeration.**

[Figure]

**Figure 6: Detailed top Khatatba Fm. time structure map of the Phiops inversion trend. Other wells have been removed for clarity.**
**See Fig. 4 for location and legend. Position of Fig. 5 is indicated.**

[Figure]

**Figure 7: Regional geoseismic section across the eastern part of the Western Desert. Location is shown in Fig. 2. The massive**
**Mubarak inversion is one of the best examples of Western Desert Santonian inversion followed by younger pulses of shortening.**
**After Bosworth et al. (2008). 4x vertical exaggeration.**

[Figure]

**Figure 8: Simplified structural map of the Black Sea modified from Tari and Simmons (2018). Location is shown in Fig. 1. Within the Black Sea itself, the depth to break-up unconformity is shown, cold colors indicating larger depth, adapted from Robinson (1997). Black lines between the Carpathians and the Black Sea correspond to major faults pre-dating the opening of the Black Sea (Krezsek et al., 2017). White triangles represent offshore mud volcanoes and red dots represent Cretaceous paleo-volcanoes (Nikishin et al., 2015a). The locations of a depth-converted regional seismic line (Fig. 10), two vintage seismic profiles (Figs. 11 and 12) and a Maykop (Oligocene to Lower Miocene) isopach map (Fig. 13) are shown by red lines.**

[Figure]

**Figure 9: Regional correlation of wells drilled on the Odessa Shelf, compiled from various sources (e.g. Gozhik et al., 2006, 2010). For location see map inset. Depth of stratigraphic tops are in meters. Note that none of the deep basins (e.g. Karkinit basin) have been penetrated to their full depth, unlike the basement highs (e.g. Kalamit high).**

[Figure]

**Figure 10: Regional-scale pre-stack depth migrated (PSDM) seismic reflection profile across the Odessa Shelf, courtesy of ION.**
**For location see Fig. 8. Note the >7500 m deep Karkinit trough in the middle of the section and the Kalamit ridge to the south of it.**
     **~5x vertical exaggeration.**

[Figure]

**Figure 11. Legacy 2D seismic reflection profile across the undrilled Gordievicha prospect, adapted from Burchell (2008). For**
     **location see Fig. 8. The position of the null-point, *sensu* Williams et al. (1989), is shown by a red star.**

[Figure]

**Figure 12: Seismic reflection evidence for post-Sarmatian inversion. For location see Fig. 8. The southward prograding Pliocene sequence above the Sarmatian (Late Miocene) unconformity (shown in blue) is clearly back-rotated. This is due to the multiple**
**episodes of inversion forming the overall structure containing the relatively small Shtormovaya field on its northern flank (e.g. Khriachtchevskaia et al., 2009). Note the gradual incorporation of the earlier Eocene folds into a much larger Miocene to Pliocene inversion anticline. Vertical exaggeration is ~6x assuming an average seismic velocity of 4 km/s.**

[Figure]

**Figure 13. Isopach map of the post-inversion Oligocene to Lower Miocene Maykop Suite in the Odessa Shelf, modified from Gozhik (2010). For location see Fig. 8. Contour intervals are in meters. Note that the thickest Maykop is not captured by the currently available well control (cf. Fig. 9). The depicted sediment entry points and the deep-water distribution patterns are entirely speculative and are shown here to highlight the stratigraphic trapping potential in the Karkinit basin.**

[Figure]

**Figure 14a. Schematic illustration of some of the possible effects of superimposed, alternating phases of extension and inversion on a rift basin. For simplicity two inversion events are shown affecting different faults, which is commonly observed in the Western Desert of Egypt but will not always be the case.**

[Figure]

**Figure 14b. Schematic illustration of a basin in which early extension is followed by multiple phases of inversion. The transect is largely based on observations made in the NW Black Sea which experienced at least 4 distinct inversion episodes.**